# Failure Study of BFRP Joints with Two Epoxy Adhesives under Hygrothermal Coupling

**DOI:** 10.3390/polym15193949

**Published:** 2023-09-29

**Authors:** Ruitao Niu, Yang Yang, Yinghao Lin, Zhen Liu, Yisa Fan

**Affiliations:** 1School of Aerospace Engineering, Zhengzhou University of Aeronautics, Zhengzhou 450046, China; niu.ruitao@163.com; 2Institute of Mechanical Engineering, Materials and Transportation, Peter the Great Saint-Petersburg Polytechnic University, Saint-Petersburg 195251, Russia; 3School of Mechanical Engineering, North China University of Water Resources and Electric Power, Zhengzhou 450045, China; 15953526518@163.com (Y.L.); fanyisa123@163.com (Y.F.); 4School of Aeronautics, Northwestern Polytechnical University, Xi’an 710072, China

**Keywords:** basalt fibre-reinforced polymer (BFRP), adhesive joint, hygrothermal degradation, failure mechanism

## Abstract

Basalt Fibre Reinforced Polymer (BFRP)-bonded structures are lightweight, high strength, economical, and environmentally friendly, which is very advantageous in the civil sector. The aim of this paper is to provide a comprehensive account of the hygrothermal degradation and failure mechanisms of BFRP-bonded structures by comparing the residual properties of two epoxy adhesive BFRP single-lap joints after ageing for 240 h, 480 h, and 720 h in an extreme hygrothermal environment with pure water at 80 °C. The hydrophilicity and thermal stability of the two adhesives were firstly compared by water absorption and Thermogravimetric Analysis (TGA) tests, and the hygrothermal degradation of the molecular chains and the reduction in T_g_ were characterised by Fourier Transform Infra-Red (FTIR) spectroscopy and Differential Scanning Calorimetry (DSC) curves. The failure strength and load-displacement curves of the two joints were then compared, and it was found that the strength and stiffness had different trends, while the paired *t*-test was used to demonstrate the correlation between the failure strength and the adhesive T_g_, as well as the difference in the failure mechanisms of the two joints caused by the water absorption rate. The analysis of macrosections and Scanning Electron Microscope (SEM) images summarised the process and reasons for the transition of the failure mode from fibre tearing to hybrid failure, and finally, the changes in elemental concentration and O/C values were analysed by Energy Dispersive X-ray Analysis (EDX), which proved that the degree of hydrolysis could not be used as a judgement of the degradation degree of the joint alone, and provided data support for the application of the BFRP-bonded structure in the humid and hot environment.

## 1. Introduction

Fibre-reinforced composite-bonded structures are one of the key ways to solve the problem of lightweight in the manufacturing industry, which has considerable application prospects in aerospace [1], automotive [2], shipbuilding [3], and wind power energy [4]. Among them, fibre-reinforced composites (FRPs) have the advantages of high specific strength to stiffness, lightweight, corrosion resistance and designability [5], replacing an increasing proportion of metal substrates in modern structures, and their usage is even considered one of the indicators of product advancement [6]. Bonding is a lightweight joining technology. Compared to traditional mechanical joining methods, adhesive bonding has good fatigue performance, vibration, noise reduction, large bearing area, no damage to structural materials, and avoidance of stress concentration [7,8]. At the same time, adhesives are generally considered more suitable for fibre-reinforced composites due to their similar polymer properties [9]. Therefore, bonded structures for fibre-reinforced composites have great advantages in terms of mechanical properties and reduced energy consumption.

The problem to be solved for fibre composite-bonded structures is their sensitivity to the environment, including factors such as temperature, humidity, UV radiation, and corrosion by acids, alkalis, and salts [10,11]. The researchers conducted ageing experiments by fabricating various geometrical types of bonded joints, thus exploring the degradation mechanism of various environmental factors. The effect of the temperature on bonded joints is commonly considered; when the temperature exceeds the glass transition temperature (T_g_), the adhesive changes from a glassy state to a highly elastic state, and the strength and stiffness are significantly reduced [12]. There is a significant difference between the coefficient of thermal expansion of the fibre and the resin matrix in FRP, and the thermal stresses generated between the two under temperature changes can cause pores, cracks, or delamination, affecting the transfer of stresses, while temperature changes can also induce creep behaviour in the material. The effect of the temperature is not always unfavourable; the post-curing phenomenon of the adhesive due to warming can improve the mechanical properties of the bonded structure [13,14]. Nguyen et al. [15] conducted ageing tests on CFRP/steel-bonded joints at different temperatures and loading conditions and found that the degradation of stiffness and strength of the joints showed a certain functional relationship with both temperature and ageing time. Akderya et al. [16] investigated the effect of thermal ageing on the tensile properties of fibre/epoxy composites at three different temperatures (18 °C, 25 °C, and 75 °C) for single-lap joints. The results showed that thermal ageing at 18 °C increased the load-carrying capacity of the joints; however, the mechanical properties tended to decrease again as the ageing temperature increased. Moisture ageing is equally unavoidable in real environments, where moisture enters the material partly in the free state in the pores and partly bound to hydrophilic groups. The mechanisms by which moisture induces degradation of the adhesive structure are water absorption expansion, hydrolysis, plasticisation, etc., but these damages have been found to be reversible after drying to a certain extent [17]. The effect of humidity on bonded joints is related to the method of bonding, as well as the surface treatment, in addition to material and time constraints [18]. Zhang et al. [19], by establishing a functional relationship between moisture concentration and local mechanical properties of epoxy adhesives, found that a high moisture concentration in the edge region of the adhesive resulted in a significant decrease in its properties, which improves the basis for improving the accuracy of numerical simulation of epoxy adhesives. Heshmati M. et al. [20] explored the effect of adhesive permeability and moisture distribution at the bonding interface on the properties of bonded joints using finite element simulation and found that there is a direct relationship between mechanical properties and moisture content.

The effect of a single environment on FRP-bonded joints is limited, and moisture–heat coupling is more in line with the reality of the environment, where the ageing rate of the FRP-bonded structures will be significantly accelerated by the interaction of the various mechanisms. You et al. [21] investigated the effect of moisture–heat coupling on the mechanical properties of nano-SiO_2_-bonded specimens and the bonding properties of CFRP steel lap joints by accelerated ageing tests, and the results showed that the water bath temperature has a great influence on the tensile and elastic properties of epoxy adhesives. Qin et al. [22] found that a single temperature, humidity, or load had a limited effect on the failure strength degradation of butt joints, whereas multifactorial coupling had a more pronounced effect. When the ageing environment was coupled with load, the ageing environment played a major role, and load accelerated the ageing rate but did not change the degradation trend caused by the ageing environment. Reyhan Ceylan et al. [23] investigated and compared the single-lap bonding joints of AA7075-T6 aluminium adherends bonded with non-reinforced as well as epoxy adhesives reinforced with different nanomaterials BNNP (Boron Nitride Nanopowder), ANP (Aluminium Oxide Nanopowder), and MWCNT (Multi-Walled Carbon Nanotubes) after exposing them to 60 °C/100% RH for 1, 2, 4, and 6 weeks. The results show that the shear strength decreases under the influence of hygrothermal ageing due to oxidation occurring as a result of water molecules entering the bonding region and that the nanomaterials reinforced bonded joints have a higher shear strength as compared to the non-reinforced bonded joints.

The failure modes of FRP-bonded joints are more complex than those of the metals [24,25] and are mainly classified according to the material damage as fibre tearing of FRP, cohesive failure of the adhesive, debonding failure of the bonding interface, and hybrid failure [26]. The joints crack at the weakest point after ageing in the hot and humid environment, which causes a change in the failure mode due to the difference in the degree of ageing of the FRP, the adhesive, and the bonding interface. At the same time, the ageing rate of joints under different humid and hot environments is not the same, and the resistance of joints bonded by different adhesives to humid and hot ageing is not the same. Therefore, it is of great significance to investigate the effects of temperature and humidity on the ageing rate and failure mode of bonded joints, as well as the mechanical properties and ageing resistance of joints bonded by different adhesives on the practical engineering applications of FRP-bonded structures.

The advantages of basalt fibre-reinforced composites (BFRP) over other FRP materials are the wide source of raw materials, lower cost, recyclability, and environmental friendliness [27], which have great potential for application in civil facilities. At present, the research on FRP-bonded joints mainly focuses on CFRP and GFRP [28,29,30,31], but the cost is an important reason for limiting its use in large quantities in the civil field, so it is worthwhile exploring the more economical BFRP. The mechanism of hygrothermal ageing of BFRP-bonded joints has not been fully elucidated. Therefore, in this paper, two epoxy adhesives, Araldite^®^2011 and Araldite^®^2014, were selected for the fabrication of BFRP single-lap joints, and ageing experiments were carried out under 80 °C pure water environment. Comparative analyses were carried out to summarise the effects of the different adhesives on the failure mechanism of the joints and to further understand the contributions and interactions of the influencing factors in the process of hygrothermal ageing so as to provide data support for the wider application of the BFRP-bonded structures in civil facilities.

## 2. Experimental Programme

### 2.1. Materials

The unidirectional fibre cloth lay-up direction in the BFRP board (Zhongdao Science and Technology Company, Jilin, China) selected for this experiment is [0/90/0/0/90/0/90], and the fibre cloth is processed by basalt fibre prepreg with ML-5417A (epoxy resin)/ML-5417B (curing agent) as the matrix, and the performance parameters are shown in Table 1. The two adhesives selected are structural adhesives widely used in the industry, namely, Araldite^®^2011 and Araldite^®^2014 (Huntsman Advanced Materials Co., Ltd., Shanghai, China). Araldite^®^2011 is a multi-purpose, two-component adhesive with high strength, low shrinkage, good toughness and dynamic load performance. Araldite^®^2014 is a thixotropic, room temperature-curing epoxy adhesive resistant to environmental corrosion. The adhesive properties are shown in Table 2.

### 2.2. Specimen Preparation

The simplest and most effective single-lap joint was chosen for the hygrothermal ageing experiments. According to the fabrication standard of the joint, ASTMD586801 [32], the geometric dimensions of the laminate are 100 mm × 25 mm × 2 mm. The geometric conditions, such as the area of the bonding area and the thickness of the adhesive, play a key role in the performance of the joint. In order to obtain the optimum bonding performance, the lap length was determined to be 25 mm, the thickness of the adhesive layer to be 0.1 mm, and the geometric dimensions of the joint are shown in Figure 1.

Single-lap joints were fabricated in a dust-free environment at a temperature of 25 ± 2 °C and a humidity of 50 ± 5%, and the surface treatment of the substrate was required to prevent the substrate surface adherents from affecting the performance of the joints [33]; the surface of the area to be bonded with BFRP was wiped with acetone to remove dust and grease before bonding, and the bonding was performed after drying for 15 min. The thickness of the adhesive layer was controlled with a glass bead with a diameter of 0.1 mm, and after bonding, it was left at room temperature for 16 h for curing.

In order to individually analyse the moisture absorption pattern of each material under an 80 °C pure water environment, dumbbell-type specimens of both adhesives were prepared for water absorption test according to the standard ASTM D638-14 [34]. The two-component glue gun was used to fill the adhesive evenly into the mould, and care was taken to prevent air bubbles from appearing in the dumbbell specimens during the filling process.

### 2.3. Experimental Design

Experimentally produced BFRP single-lap joints bonded by two adhesives were subjected to hygrothermal ageing in a pure water environment at 80 °C. Three experimental groups were set up for ageing for 240 h, 480 h, and 720 h for each type of joints, and an unaged reference group and three joints were placed in each group. BFRP joints were placed in the drying chamber for drying, to remove the water absorbed by the adhesive layer during the curing process; after drying, the joints were ageing-processed, grouped into pure water, and finally, put into the temperature and humidity-coupling environment box (Weiss Equipment Experimentation Co., Ltd., WS–1000, Dongguan, China) to achieve the corresponding ageing time, after which they were removed and then left at room temperature for 8 h for quasi-static tensile test, and the stretching rate was selected to be 2 mm/min, as shown in Figure 2.

Water absorption tests were carried out on BFRP panels and adhesive dumbbell-type specimens, three specimens were taken for each material, and finally, the data obtained were averaged. The specimens were weighed regularly at 24 h intervals using an analytical balance with an accuracy of 0.1 mg, and the water absorption rate was calculated by comparing the initial mass. Before weighing, absorbent paper was used to remove the surface moisture, and the weighing process was no more than 10 min to avoid the external environment affecting the experimental results. The water absorption rate was calculated by Equation (1):(1)$$M_{t}=\frac{W_{t}-W_{0}}{W_{0}}\times 100\%$$
where *M_t_* denotes the water absorption at time *t*; *W_t_* denotes the mass at time *t,* and *W*_0_ denotes the initial mass.

In addition to the water absorption of the material, thermal stability is one of the basis for adhesive selection, and its impact on the mechanical properties of the joint is also very important. Thermogravimetric analysis (TGA) was carried out for both adhesives; the tests were carried out in purged N_2_ between room temperature (RT) and −800 °C, with a heating rate of 5 °C/min and sample weights of 25–45 mg.

In order to investigate the mechanism of hygrothermal ageing of joint materials at the molecular level in depth, Fourier Transform Infra-Red tests were carried out on two types of adhesives, where groups of adhesive samples were measured under N_2_ ambient conditions in the wavelength range of 500–4000 cm^−1^. The samples were approximately 10 mg, and each spectrum was scanned 32 times with a spectral resolution of 4 cm^−1^, and the FTIR spectra at each ageing node were recorded. At the same time, the T_g_ changes in the two adhesives were obtained by using a differential scanning calorimeter (Mettler Toledo, DSC3+, Greifensee, Switzerland), which was also tested in N_2_ environment with a heating rate of 5 °C/min and a temperature range of −50–150 °C. During the test, it is necessary to heat up twice. The first time, it is carried out to remove the thermal history and the second test result is taken as T_g_ [35].

The ageing resistance of the adhesive is one of the key factors in the durability of BFRP joints, and it also determines the failure mode. The failure mode of a joint can be observed from the appearance of the joint section, and changes in the failure mode directly reflect the evolution of the degradation of the various components of the joint. The characteristic areas of the section were selected for SEM and EDX analysis to further summarise the causes of joint fracture and failure mechanisms from the micro-morphology and element distribution. Figure 3 shows the overall design framework of the experiments in this paper.

## 3. Results and Discussion

### 3.1. Moisture Absorption Characteristics

Moisture will lead to hydrolysis of the material when it enters the material, and the movement of moisture will make the micropores in the material expand or even induce cracks; all of these adverse effects will reduce the mechanical properties of the material. Water absorption is a reference value that can directly reflect the influence of water on the material. Figure 4 shows the experimentally measured water absorption of BFRP panels in 720 h. Fick’s law is a widely used model for describing the diffusion of substances, and the water absorption curve is obtained by fitting the water absorption rate with Fick’s law and the fitting equation is shown in Equation (2) [36]:(2)$$M_{t}=M_{\infty }\times \left[1-\frac{8}{\pi^{2}}\overset{\infty }{\underset{n=0}{\sum }}\frac{1}{{\left(2n+1\right)}^{2}}\exp\left(-\frac{{\left(2n+1\right)}^{2}\pi^{2}D}{h^{2}}t\right)\right]$$
where *M_t_* is the water absorption at moment *t*; *M*_∞_ is the saturated water absorption; *D* is the diffusion coefficient, and *h* is the thickness of the sample. Analysing the water absorption curve, it can be found that the water absorption rate decreases rapidly as the water absorption tends to lead to saturation, and Fick’s law fits the change in water absorption rate very well at the early stage. However, when the diffusion tends to be saturated at the later stage, the water absorption rate test value gradually decreases, but there is a significant deviation from the fitted curve, and the actual water absorption rate is larger than the fitted value, indicating that the water absorption mechanism in the material is not purely free diffusion, and there are other water absorption mechanisms to promote the material to further absorb water after diffusion tends to be saturated. Moisture exists in the material in two states, a free state and a bound state; as explained in Figure 4, the free water mainly fills the cavities in the resin matrix and the gaps between the fibre and the resin, while the bound water combines with the hydrophilic groups in the material by hydrogen bonding. Although the amount of bound water is much lower than the free water, it is the formation of bound water that causes the actual water absorption to be greater than the fitted value at a later stage. 

Figure 5 shows the water absorption of two epoxy adhesives, and the trend and phenomenon of the fitted curves are consistent with that of BFRP, indicating that the water absorption mechanism of the adhesive is similar to that of BFRP. The total water content in the adhesive is related to the water absorption time, but the water content in different areas of the adhesive at the same moment is also limited by time. Moisture enters in the form of diffusion from the edge of the adhesive and gradually moves to the centre of the adhesive, so the edge of the adhesive is more severely affected by moisture ageing than the centre area, and there are more cracks and pores, from which it is deduced that failure in the bonded area tends to be transmitted from the edge of the adhesive to its centre. A comparison of the water absorption of the two adhesives shows that Araldite^®^2011 is more hydrophilic than Araldite^®^2014.

### 3.2. TGA-DTG Test

In order to understand the thermal stability properties of the two epoxy adhesives and the effect of moisture and heat ageing on this property, unaged and aged 720 h samples of the adhesives were extracted for TGA testing, while in order to accurately observe the range of weight loss and the rate of weight loss, the thermal weight loss curves were differentiated and the DTG curves were obtained, as shown in Figure 6. When the temperature rises to 120 °C, the rapid release of water and volatiles from the adhesive leads to the formation of a small weight loss peak near this temperature [37], whereas Araldite^®^2011 has a larger peak due to its greater water absorption than Araldite^®^2014. With temperatures in the range of 200–600 °C, the adhesive pyrolysis, the quality of the rapid reduction in the temperature range, the adhesive has two weight loss peaks; the size of the peak is related to the composition of the adhesive. When the temperature reaches 800 °C, the residual rate of Araldite^®^2011 is not higher than 10%, while the residual rate of Araldite^®^2014 is greater than 40%, indicating that Araldite^®^2014 is more thermally stable than Araldite^®^2011. The difference between the curves after hygrothermal ageing for 720 h and the unaged curves is not significant, but there is a small increase in the final residual rate of the aged samples of both adhesives, from which it is deduced that a small amount of more thermally stable substances are generated by the reaction during hygrothermal ageing.

### 3.3. FTIR Test

The FTIR test results of the adhesive are shown in Figure 7. The samples were observed in FTIR spectra with the assignment of each characteristic absorption band refer to Table 3 [38]. 

The whole measurement range during this analysis can be divided into two parts from 1330 cm^−1^. A section of 1330–4000 cm^−1^ belongs to the characteristic frequency region, where the absorption peaks are mainly generated by the stretching vibration of specific functional groups, and the 500–1330 cm^−1^ section belongs to the fingerprint region, where the absorption peaks are numerous and complex, mainly relating to the vibration of the single bond, the hydrogen-containing group, and the C skeleton. In order to better study the spectral changes during ageing, the intensity of the band at 1508 cm^−1^, which is considered to be the most stable structure in polymeric materials and is essentially uninvolved in hygrothermal ageing, was used as a benchmark [39]. The hydroxyl group (OH) corresponding to the wave number 3100–3325 cm^−1^ is closely related to water absorption, and the absorption strength increases significantly with ageing because the high temperature in the early ageing stage accelerates molecular movement; the adhesive absorbs water rapidly, and a strong hydrogen bond is formed between the water molecule and the resin network [40]. As water absorption approaches saturation, water molecules diffuse into the material and remain in the free volume to react with certain functional groups within the material [41], which are modified by hydrolysis or oxidative degradation. With the increase in ageing time, the absorption peaks of the carbonyl group (1648 cm^−1^) and ether bond (1242 cm^−1^) increased, while the absorption peak of the ester group (1736 cm^−1^) decreased in parallel, which is presumed to be the localized hydrolysis and breakage of the ester group in the polymer molecular chain in the epoxy resin adhesives to form new carbonyl and ether bonds.

Since Araldite^®^2011 has a higher water absorption than Araldite^®^2014, the changes in the individual absorption peaks are more pronounced. The corresponding wave number band for the epoxy group is 914 cm^−1^, and although the absorption peaks of the epoxy group are attenuated in both adhesives, the absorption peaks of Araldite^®^2011 are higher than those of Araldite^®^2014 in comparison, a phenomenon suggesting that the density of the epoxy molecular chain in Araldite^®^2011 may be reversibly greater in the same sample volume.

### 3.4. DSC Test

Changes in polymer chain length, functional groups, cross-link density, and molecular weight can alter the T_g_ of the adhesive [42]. There are two main mechanisms affecting T_g_ during hygrothermal ageing: (i) hydrolysis and pyrolysis cause polymer chains to break and cross-link density to decrease, resulting in a decrease in T_g_; (ii) the formation of hydrogen bonds between the bound water and the hydrophilic groups, resulting in an increase in T_g_. Comparing the T_g_ of the two adhesives, the observation of Figure 8 shows that with the increase in ageing time, the T_g_ of both adhesives shows a decreasing trend, which proves that the degradation of the two adhesives has occurred to different degrees. The decrease in T_g_ causes the adhesive to change more easily from a glassy to a highly elastic state, which reduces the load-bearing capacity of the bonded joint. From the water absorption results, it can be seen that Araldite^®^2011 is more hydrophilic than Araldite^®^2014, as it contains more bound water and forms more hydrogen bonds, resulting in a smaller decrease in T_g_ for Araldite^®^2011 than for Araldite^®^2014.

### 3.5. Mechanical Properties

The failure strengths of single-lap joints with two different adhesives after hygrothermal ageing are shown in Figure 9; the coloured areas are error bands. From the failure strength data, both before and after ageing, the mechanical properties of Araldite^®^2011-bonded joints are superior to Araldite^®^2014-bonded joints, suggesting that Araldite^®^2011 adhesives provide better adhesion properties and bonding interfaces to the joints. The failure strength of the joint continues to decrease with the ageing time. In the early stage of ageing, the high temperature promotes rapid water absorption and degradation of the joint material, and the failure strength of the joint is greatly reduced; the water absorption tends to be saturated in the later stage, and the effect of humidity and heat ageing slows down. After ageing for 720 h, the residual strength of Araldite^®^2011-bonded joints decreased by 26.18%, and Araldite^®^2014-bonded joints decreased by 38.95% compared to the unaged joints. The comparison shows that Araldite^®^2011-bonded joints have better ageing resistance than Araldite^®^2014-bonded joints, which is attributed to the fact that Araldite^®^2011 adhesive forms more bound water after absorbing water, and the hydrogen-bonding enhances the cross-linking of the polymer chains, which also reduces the decrease in T_g_.

To further investigate whether there is any consistency between the failure strengths and T_g_ decrease rates of the two joints at the three ageing nodes, they were subjected to paired *t*-tests between them in pairs, and the results are shown in Table 4. When the *p*-value is lower than 0.05, it means that the data exhibit significant variability; according to the results of the *p*-value, there is no variability between the failure strength decline rate and the T_g_ decline rate of the joint, which proves that the T_g_ of the adhesive is one of the key factors affecting the failure strength of the joint. However, the failure strength decline rate and T_g_ decline rate of the two joints showed differences, suggesting that there is a difference in the failure mechanism of the two joints in hygrothermal environments, which is attributed to the different water absorption rates of the adhesives. Larger values of Cohen’s *d* indicate more significant variability, and a comparison of Cohen’s *d* values reveals that the difference in the failure strength decline rate is greater than the difference in the T_g_ decline rate. This is because the difference in water absorption not only affects the adhesive but also acts on the bonding interface of the joint.

The load-displacement curves show the whole failure process of the joints, and it can be found from Figure 10 that the failure of the joints is sudden and completely loses the load-carrying capacity in a very short period of time, so even a slight decrease in the failure load poses a great risk to the bonded joints. The trend of the load-displacement curve is consistent with the failure strength, and the reduction in the failure displacement is mainly concentrated in the pre-ageing period. It is worth noting that as the ageing time increases, the slope of the load-displacement curve increases and then decreases, indicating that the stiffness of the joints is not monotonically reduced like the strength, and the increase in the stiffness in the early stage also proves that the joints absorb water to form hydrogen bonding to increase the cross-linking density. However, with the prolongation of the ageing time, the continuous hygrothermal degradation of the joints will also cause the stiffness to decrease continuously.

### 3.6. Failure Mode

Observation of the joint failure section and SEM electron microscope scanning of the characteristic region were carried out to analyse the evolution of the failure mode in the bonding region, which, in turn, enabled the weak position of the joint and the extension of the failure to be deduced, as shown in Figure 11. The failure mode of the joints in the unaged condition was mainly in fibre tearing, demonstrating that the bond strength of the adhesive met the requirements of the material used. However, the failure mode of the joints after hygrothermal ageing becomes a mixed failure, with obvious adhesive layer cohesion phenomenon and localised interface failure, indicating that the weak point leading to the failure of the joints has changed from the BFRP to the adhesive and bond interface. Because the water absorption of the adhesive is much greater than that of BFRP, and because the moisture enters the adhesive pores and expands to induce microcracks, the hygrothermal degradation of the adhesive is much more serious than that of BFRP, leading to the adhesive layer’s cohesion failure. The bond interface relies on the adhesive and BFRP to form chemical bonding and mechanical occlusion to provide adhesion. When moisture enters the bond interface from the bonding line, it will destroy the connection between the adhesive and the BFRP, and the load is transferred between the two materials due to the weakening of the interfacial connection, which eventually results in interfacial debonding failure. The evolution of the failure mode of the joint after hygrothermal ageing is shown in Figure 12.

Exposed fibre filaments and fibre breaks in the BFRP can be observed from the SEM images of the characteristic areas. The characteristic areas selected for the Araldite^®^2014-bonded joints aged for 720 h are on the surface of the adhesive, and it can be observed that rough folds are formed in the section of the adhesive after ageing, which is a result of the plasticising effect of hygrothermal ageing on the adhesive, as well as the softening of the adhesive due to the reduction in T_g_. A comparison of the sections of the two adhesive joints shows that the Araldite^®^2011-bonded joints are rougher than the Araldite^®^2014-bonded joints, suggesting that Araldite^®^2011 has greater toughness than Araldite^®^2014 at fracture.

### 3.7. EDX Test

Sections of two joints aged for 240 h and 720 h were selected for EDX testing to analyse changes in the apparent concentrations of various elements. Figure 13 shows the types and distribution of the elements contained on the surface, from which it is observed that element C is the element with the highest apparent concentration and is present in both the fibre and the resin, but it is mainly concentrated in the resin. O is also one of the elements with high concentration, and it is distributed throughout the observation area, but the distribution characteristics are opposite to those of C, with a higher concentration in the fibres. The concentrations of Si and Fe are relatively low and are clearly concentrated in the basalt fibres.

The apparent concentration and weight percentage of each element are listed in Table 5. After ageing, the content of element C decreased considerably, proving that the polymer chains underwent severe degradation in hot and humid environments. O/C can indicate the degree of hydrolysis; the O/C of the two adhesive joints showed different degrees of increase, and the comparison found that the increase in Araldite^®^2011 adhesive joints was significantly higher than that of Araldite^®^2014 adhesive joints, indicating that the hydrolysis of Araldite^®^2011 adhesive joints was more serious, which corresponded to the greater water absorption of Araldite^®^2011. However, it is clear from the failure strength that Araldite^®^2011-bonded joints have a higher load-carrying capacity and better ageing resistance, suggesting that the degradation of bonded joints cannot be judged solely on the basis of water absorption and hydrolysis of the material but that joint performance is also influenced by such factors as T_g_ and the bonding interface.

## 4. Conclusions

BFRP-bonded joints have great potential for use in civil installations due to their economical and environmentally friendly advantages. In this paper, BFRP-bonded joints made of two different epoxy adhesives were selected and aged under 80 °C/pure water environment for 240 h, 480 h, and 720 h to study the hygrothermal degradation mechanism and failure process of the joints. These experiments first characterised the ageing of the adhesive; the water absorption of an adhesive does not fully comply with Fick’s law of diffusion because water enters the interior of the adhesive as both free and bound water. FTIR spectroscopy showed that the polymer chains were broken after hygrothermal degradation, which, in combination with the plasticising effect of moisture, led to a decrease in T_g_ in the DSC curves. Comparing the two adhesives, although Araldite^®^2014 is more thermally stable than Araldite^®^2011, the greater hydrophilicity of Araldite^®^2011 results in the formation of more hydrogen bonds upon water absorption, resulting in a smaller reduction in its T_g_.

The failure strength of the joints was further analysed in relation to the degradation of the adhesive, and the failure strength of the two joints decreased to different degrees after ageing, and the mechanical properties and ageing resistance of the joints bonded with Araldite^®^2011 were better than those of the joints bonded with Araldite^®^2014, suggesting that Araldite^®^2011 is more suitable for bonding BFRP. The paired *t*-test shows that the decrease in joint failure strength is related to the decrease in adhesive T_g_ and that there is a difference in the failure mechanism of the two joints due to the different water absorption rates of the adhesive, which is not only reflected in the T_g_ decrease rate but also acts on the adhesive interface. The load-displacement curves show the failure process of the joints, where the joint stiffness first increases due to the increase in hydrogen bonding and then decreases due to degradation and plasticisation. 

The change in the failure mode explains the reason for triggering the joint failure; the adhesive is more sensitive to the hot and humid environment than the BFRP; the high temperature promotes the moisture to enter the pores of the adhesive to expand and induce the micro-cracks, and at the same time, the moisture will also weaken the bond at the adhesive interface, and the failure in some areas of the joints is shifted from the BFRP to the adhesive and adhesive interface so that the failure mode of the joints is shifted from fibre tear to hybrid failure. The changes in section element concentrations and O/C values obtained by EDX tests demonstrate that Araldite^®^2011 has a greater degree of hydrolysis than Araldite^®^2014, but the degree of degradation in its bonded joints is lower, suggesting that water absorption and hydrolysis alone cannot be used to assess the ageing resistance of the joint. The degradation of the joint is also related to the adhesive T_g_ and bonding interface, and the failure mechanism of this part has been proven to be different due to the difference in water absorption.

## Figures and Tables

**Figure 1 polymers-15-03949-f001:**
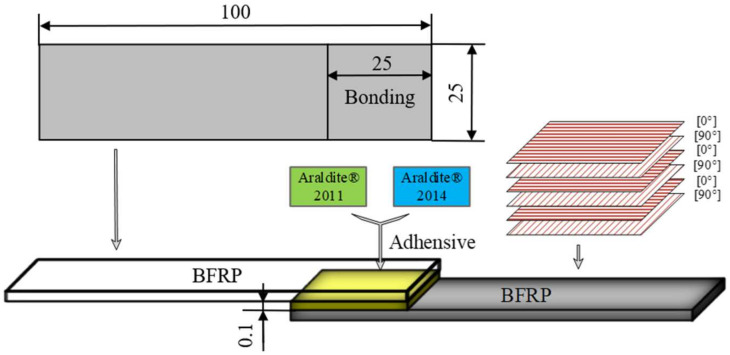
Single-lap joint geometry (mm).

**Figure 2 polymers-15-03949-f002:**
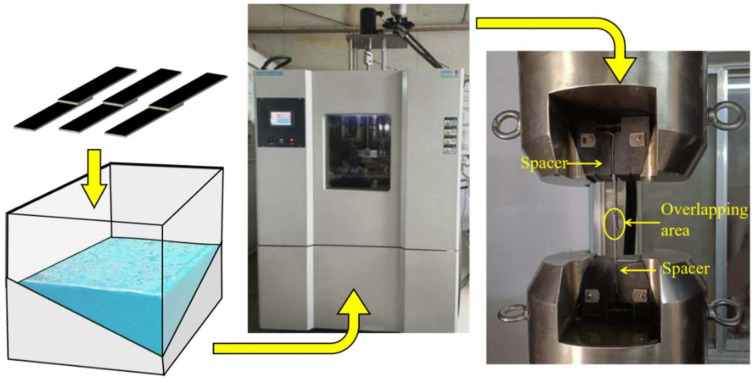
Experimental steps for joints.

**Figure 3 polymers-15-03949-f003:**
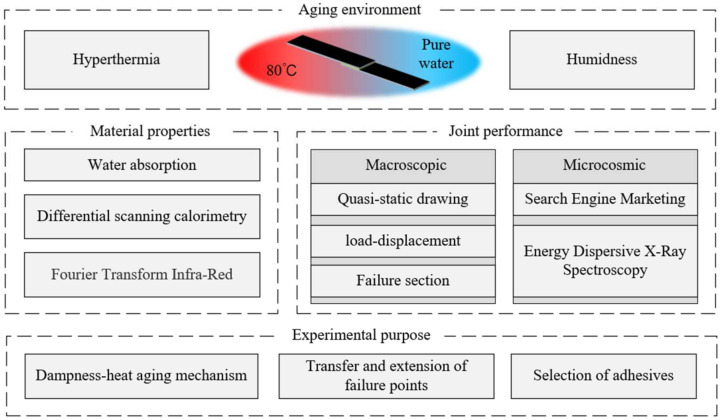
General design framework of this experiment.

**Figure 4 polymers-15-03949-f004:**
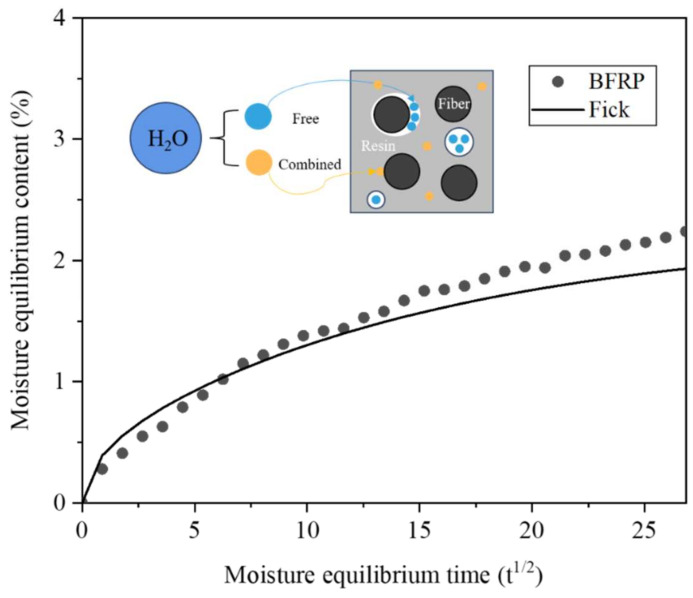
Water absorption of BFRP.

**Figure 5 polymers-15-03949-f005:**
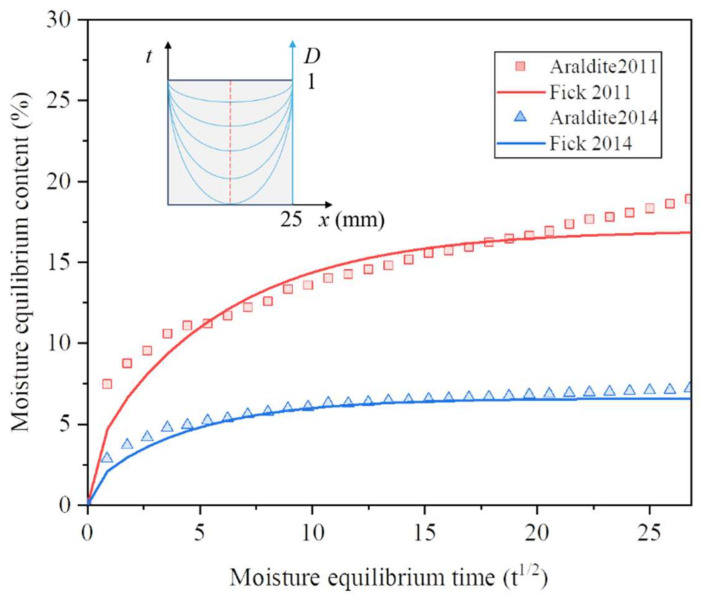
Water absorption of adhesives.

**Figure 6 polymers-15-03949-f006:**
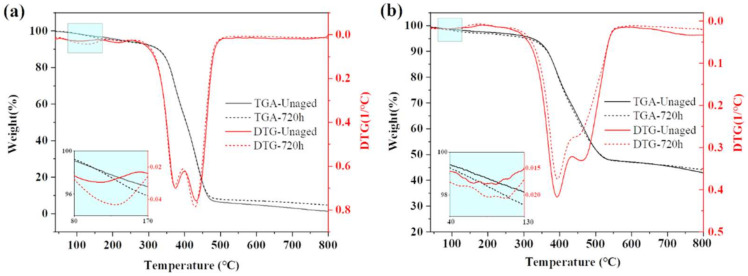
TGA-DTG test curves for two adhesive samples: (**a**) Araldite^®^2011; (**b**) Araldite^®^2014.

**Figure 7 polymers-15-03949-f007:**
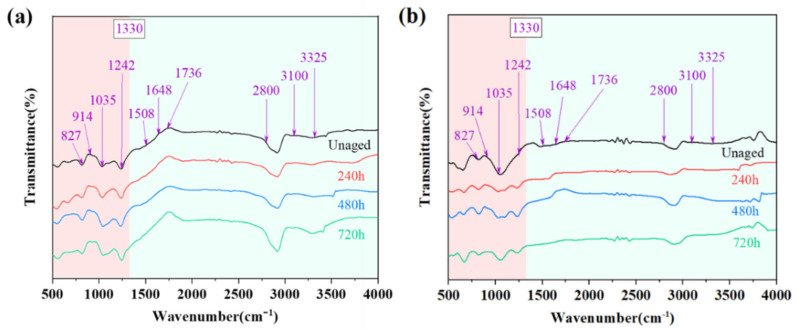
Results of FTIR performed on adhesive samples: (**a**) Araldite^®^2011; (**b**) Araldite^®^2014.

**Figure 8 polymers-15-03949-f008:**
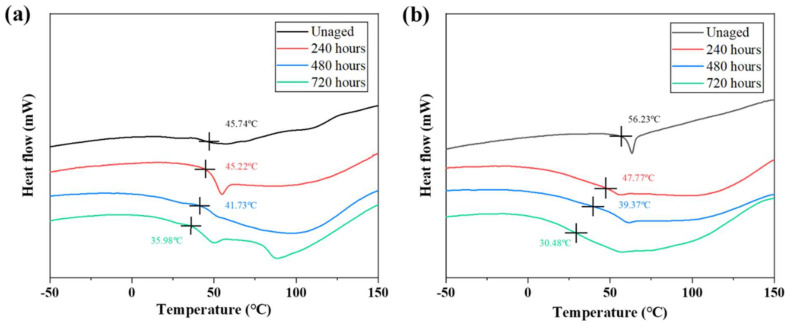
Glass transition temperatures of two adhesives: (**a**) Araldite^®^2011; (**b**) Araldite^®^2014.

**Figure 9 polymers-15-03949-f009:**
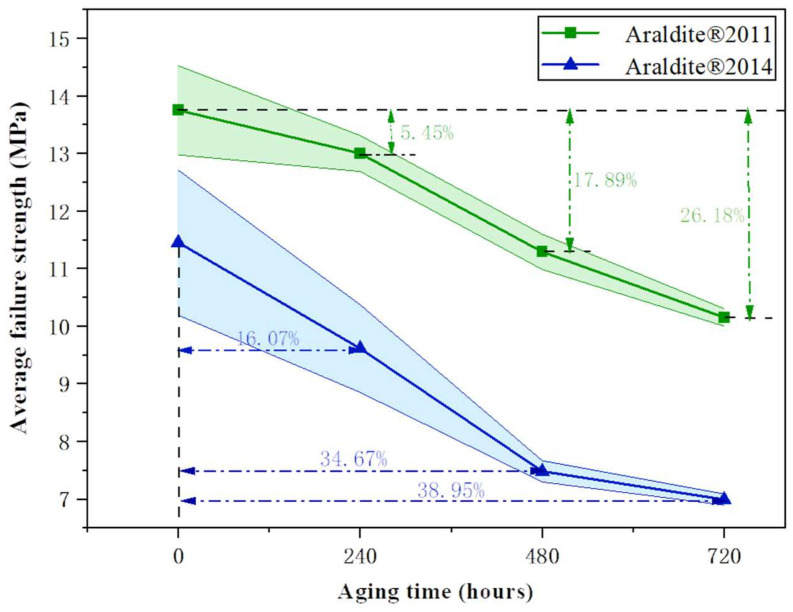
Failure strength of bonded joints.

**Figure 10 polymers-15-03949-f010:**
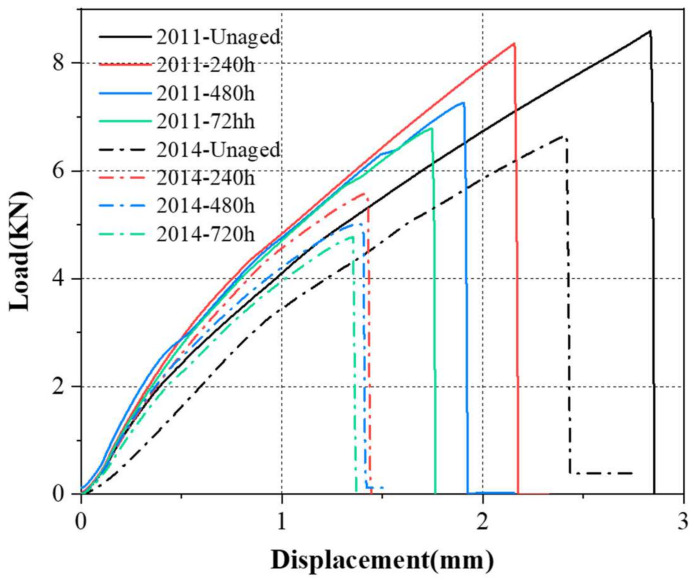
Load-displacement curves for bonded joints.

**Figure 11 polymers-15-03949-f011:**
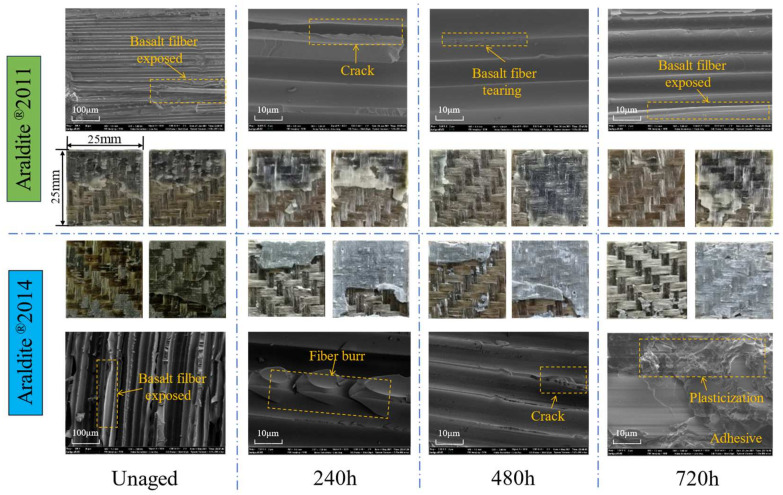
Sections and SEM images of two bonded joints.

**Figure 12 polymers-15-03949-f012:**
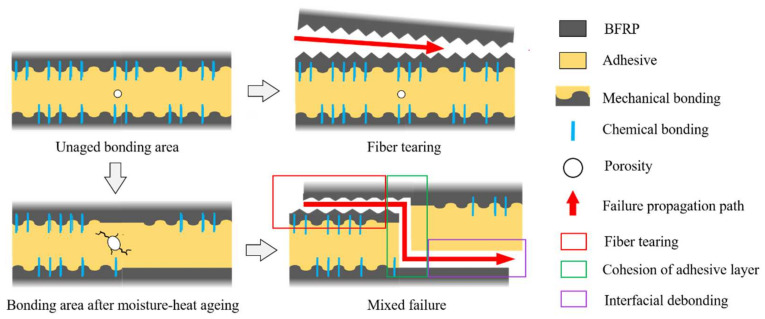
Evolution of failure modes in bonded joints.

**Figure 13 polymers-15-03949-f013:**
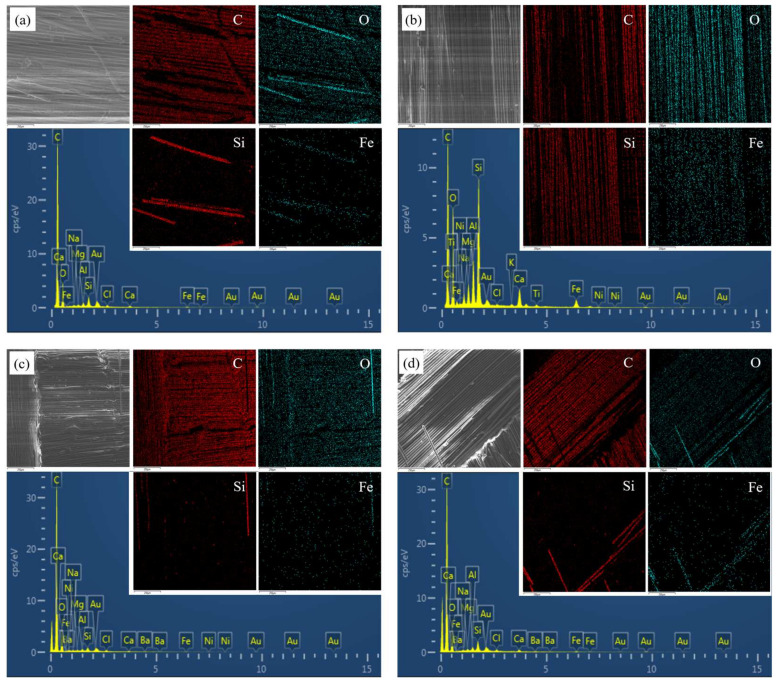
EDX test results: (**a**) Araldite^®^2011-bonded joints aged for 240 h. (**b**) Araldite^®^2011-bonded joints aged for 720 h. (**c**) Araldite^®^2014-bonded joints aged for 240 h. (**d**) Araldite^®^2014-bonded joints aged for 720 h.

**Table 1 polymers-15-03949-t001:** Performance parameters of BFRP.

ML-5417A/ML-5417B Epoxy Resin		Basalt Fibre Unidirectional Fabric	
Cure condition	25 °C × 24 h + 100 °C × 3 h	Surface density/(g·cm^−2^)	300
		Tensile strength/(MPa)	2100
Epoxy value/(g/ep)	165–175	Young’s modulus/(GPa)	105
25 °C density/(g·cm^−3^)	1.10–1.20	Elongation/(%)	2.6
Tensile modulus/(MPa)	2800–3200	Nominal thickness/(mm)	0.115
T_g_ (°C)	110–125	Single fibre size/(μm)	13

**Table 2 polymers-15-03949-t002:** Parameters of Araldite^®^2011 and Araldite^®^2014.

	Araldite^®^2011	Araldite^®^2014
Young’s modulus (GPa)	1.65	4
Shear modulus (GPa)	0.2	1.2
Density/(kg·m^3^)	1.15	1.6
Poisson’s ratio	0.43	0.33

**Table 3 polymers-15-03949-t003:** Assignments of the characteristic absorption bands observed in the FTIR spectra for the reference adhesive specimens.

Wavenumber	Assignment
cm^−1^	
3325	–OH, –NH stretching
3100–2800	Alkyl groups (–OH, –CH_2_) stretching
1736	Ester link (–(C=O)–O)
1648	Carbonyl group (C=O)
1606, 1581	Quadrant stretching of the benzene ring
1508	Semicircle stretching of p-disubstituted benzene
1452	C-H bending of aliphatic groups
1296	Twisting mode of –CH_2_– groups
1242	Stretching mode for aromatic ether
1180	C–C stretching of the bridge carbon atom between two p-phenylene groups
1082, 1035	Stretching of the trans forms of the ether linkage
914	Epoxy functional group
827	p-phenylene groups

**Table 4 polymers-15-03949-t004:** Results of paired *t*-test.

	*t*	*p*	Cohen’s *d*
A and B	−7.418	0.018	4.283
C and D	−6.366	0.024	3.676
A and C	4.011	0.057	
B and D	−0.111	0.922	

A: Araldite^®^2011 adhesive joint failure strength decline rate. B: Araldite^®^2014 adhesive joint failure strength decline rate. C: Araldite^®^2011 adhesive T_g_ decline rate. D: Araldite^®^2014 adhesive T_g_ decline rate.

**Table 5 polymers-15-03949-t005:** Concentrations of elements in the observation area.

Element	a		b		c		d	
	Apparent Concentration	wt%	Apparent Concentration	wt%	Apparent Concentration	wt%	Apparent Concentration	wt%
C	121.40	74.79	53.56	57.35	137.73	75.28	113.39	75.57
O	24.35	20.97	40.23	29.18	24.41	21.60	21.26	20.36
Si	3.03	0.91	16.76	5.42	1.20	0.43	2.86	0.95
Fe	0.63	0.23	4.26	1.58	0.28	0.12	0.84	0.34
O/C	0.20		0.75		0.18		0.19	

a: Araldite^®^2011-bonded joints aged for 240 h. b: Araldite^®^2011-bonded joints aged for 720 h. c: Araldite^®^2014-bonded joints aged for 240 h. d: Araldite®2014-bonded joints aged for 720 h.

## Data Availability

The data presented in this study are available on request from the corresponding author.

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
