# Peer review of "Failure Study of BFRP Joints with Two Epoxy Adhesives under Hygrothermal Coupling"

_polymers, 2023, doi:10.3390/polym15193949_

Round 1

Reviewer 1 Report

The manuscript discussed about BFRC (basalt fiber reinforced composites). Some questions are raised during reading this paper, for instances:

1.      The title should represent only two conditions, namely the one bonded by Araldite® 2011 and another one bonded by Araldite® 2014. Your title was unclear, only 2 types of adhesives were applied.

2.      I didn’t agree with keyword of basalt fiberboard. Fiberboard usually made of wood fiber using wet process without adhesive or dry process using adhesive. In this study the fiber was not in this terminology.

3.      Abbreviation of BFRP should appear in abstract.

4.      Please make clear about HIGROTHERMAL and HIDROTHERMAL. These two were still interchangeable in text. Please write carefully!

5.      The study would be interesting if the authors add statistical analysis which comparing BFRP bonded by Araldite® 2011 and the one bonded by Araldite® 2014, for instances using paired t test.

6.      Therefore, in the discussions the authors could add the statistical results, not only presenting data from the experiment.

7.      Fig.11 the scale was missing. Please complete them!

8.      Table 4 should show ratio of C/O since this is an indicative of the degree of hydrolysis.

9.      What is your reason comparing two conditions of Araldite? Viz, 2011 and 2014? Is it influence in practical implication? These circumstances were missing in the text.

The conclusion was not concise. Please make in a paragraph not point by point

Author Response

Thank you for your work in reviewing the article, please see the attachment for response to the review comments.

Reviewer 2 Report

In the submitted manuscript entitled Failure Study of BFRP Joints with Different Epoxy Adhesives Under Hygrothermal Coupling, the authors discussed the behaviour and failure mechanism of basalt fibre reinforced polymer (BFRP) joints with a comparison between the two adhesives under extreme hygrothermal environments. Overall, the manuscript can be interesting and can be reconsidered for publication despite the actual version requiring considerable modification. Please address the following comments

·       The abstract does not summarize the contents clearly. The abstract does not provide a suitable lead-in to the aims and objectives.

·        The authors should define the acronyms first such BFRP, FTIR, EDX...

·        In the Introduction section, state of the art novelty should be exposed clearly. There are some important general references about the subject of the paper which have been omitted in the bibliography review and that could be useful for this paper such as: - Hygroscopic effects on the penetration-resistance behavior of a specially orthotropic CFRP composite plates (10.1016/j.compstruct.2017.06.032).

·       According to curve presented in Fig 4. The water absorption is still not saturated, why the lager time used is 720h.

·       The results are not clearly stated. The results are not presented clearly and analyzed appropriately, more details about the moisture absorption characteristics explanation should be added authors invited to add some previous work where this phenomenon was explained with details. Also, authors can discuss with more details the correlation between the concentrations of elements in the observation area, the water absorption and the failure mechanism.

·        In figure 10, two curves can be merged to show both effects of hygrothermal and the used epoxy.

·       The summary is not clear. It does not give a concise statement of content. The conclusion must contain the proof of verified novelty with validation.

The manuscript contains some typos which should be corrected. Check the text for clarity, grammar, and syntax throughout.

Author Response

(The authors gave the same response as above.)

Round 2

Reviewer 1 Report

This manuscript has been fulfilled the standard and accepted in the present form. Congratulations!